# Single-Shot Multi-Frame Imaging of Femtosecond Laser-Induced Plasma Propagation

**DOI:** 10.3390/ma16083264

**Published:** 2023-04-21

**Authors:** Tianyong Zhang, Baoshan Guo, Lan Jiang, Tong Zhu, Yanhong Hua, Ningwei Zhan, Huan Yao

**Affiliations:** 1Laser Micro/Nano Fabrication Laboratory, School of Mechanical Engineering, Beijing Institute of Technology, Beijing 100081, China; zhty2019@126.com (T.Z.); gbs@bit.edu.cn (B.G.); tongzhubit@bit.edu.cn (T.Z.); huayh21@163.com (Y.H.); rabbit.ningwei@gmail.com (N.Z.); heyalex@126.com (H.Y.); 2Yangtze Delta Region Academy of Beijing Institute of Technology, Jiaxing 314000, China; 3Beijing Institute of Technology Chongqing Innovation Center, Chongqing 401120, China

**Keywords:** femtosecond laser, single-shot multi-frame imaging, wavelength polarization multiplexing, laser-induced plasma

## Abstract

Single-shot ultrafast multi-frame imaging technology plays a crucial role in the observation of laser-induced plasma. However, there are many challenges in the application of laser processing, such as technology fusion and imaging stability. To provide a stable and reliable observation method, we propose an ultrafast single-shot multi-frame imaging technology based on wavelength polarization multiplexing. Through the frequency doubling and birefringence effects of the BBO and the quartz crystal, the 800 nm femtosecond laser pulse was frequency doubled to 400 nm, and a sequence of probe sub-pulses with dual-wavelength and different polarization was generated. The coaxial propagation and framing imaging of multi-frequency pulses provided stable imaging quality and clarity, as well as high temporal/spatial resolution (200 fs and 228 lp/mm). In the experiments involving femtosecond laser-induced plasma propagation, the probe sub-pulses measured their time intervals by capturing the same results. Specifically, the measured time intervals were 200 fs between the same color pulses and 1 ps between the adjacent different. Finally, based on the obtained system time resolution, we observed and revealed the evolution mechanism of femtosecond laser-induced air plasma filaments, the multifilament propagation of femtosecond laser in fused silica, and the influence mechanism of air ionization on laser-induced shock waves.

## 1. Introduction

Femtosecond laser is characterized by ultra-high peak power (>10^12^ W·cm^−2^) and ultra-short pulse duration (10^−15^ s). It is likened to a light knife that can be used to process any variety of materials. In contrast to traditional processing, femtosecond laser processing offers rapid processing speed, precise control, and exceptional three-dimensional processing capabilities. Currently, it is playing an unparalleled role in laser additive and subtractive manufacturing [1]. However, revealing the reaction mechanism in femtosecond laser processing has always been a challenge in laser manufacturing [2,3]. The interaction between laser and material involves many nonlinear, non-equilibrium ultrafast physical and chemical processes [4]. Although the pump-probe shadowgraph technique can achieve femtosecond time scale resolution [5], it cannot achieve continuous imaging of the laser–material interaction process. Therefore, the single-shot ultrafast multi-frame imaging technology based on femtosecond laser has developed rapidly [6,7,8], and it has been gradually applied to the observation of femtosecond laser propagation [9], laser-induced plasma expansion [10], phonon vibration [11], and cell imaging [12]. How to provide an imaging method with higher imaging resolution and more imaging frames has become the goal of many researchers that expect to apply the technology to the observation of various transient processes. For example, the sequentially timed all-optical mapping photography (STAMP) can get six frame images with a high pixel resolution (450 × 450 pixels), while its frame interval reaches 190 fs, and the propagation of plasma and lattice vibrational waves was observed [11,13]. However, its imaging frame number and frame interval were both limited by the degree of spectral broadening and dispersion. Therefore, the STAMP system utilizing spectral filtering increases the number of imaging frames greatly (to 25 frames), but sacrifices the spatial and temporal resolution. The femtosecond laser-induced filament propagation inside the glass can hardly be observed clearly [14,15,16]. Therefore, for this optical imaging based on spectral stretching, the imaging frame rate and the number of frames mainly depend on the degree of spectral dispersion. With the increase in the number of frames, the luminous flux decreases sharply, resulting in insufficient imaging clarity [17], and the system is unable to capture more subtle transient processes. Compressed spectral imaging combining streak cameras and spectral compression algorithms has attracted much attention [18]. The trillion-frame-per-second compressed ultrafast photography (T-CUP) can capture a dynamic scene with spatial dimensions of 450 × 150 pixels and a sequence depth of 350 frames in a single camera exposure [9]. To improve the temporal resolution of imaging, the fusion of pulsed time-domain broadening and compressed spectroscopy shows a good development potential, compressed ultrafast spectral-temporal (CUST) photographic technique, enabling both a large frame number (60 frames) and a temporal interval of 260 fs [19]. In addition, chirped spectral mapping ultrafast photography (CSMUP) achieves a huge increase of 250 frames and a spatial resolution of 833 nm, while its temporal resolution is 4 ps [20]. These techniques greatly expand the depth of the imaging sequence. However, as mentioned above, the broadening of the spectrum will inevitably lead to the reduction of the energy of each spectrum and the insufficient luminous flux allocated to each image, resulting in the weakening of the image. Continuously increasing the pulse energy will inevitably cause damage to the optical components, resulting in some nonlinear phenomena [21], and the complex algorithm also brings difficulties to image restoration with a high signal-to-noise ratio [22].

By separating the probe pulses equally and imaging them in different frames, stable imaging quality and high spatial resolution can be provided for the observation of femtosecond laser processing. For example, all-optical coaxial framing photography observes and calculates the ultrafast electron dynamics process of a laser-ablated aluminum foil [23]. Single imaging based on the spatiotemporal division of femtosecond laser pulses observes laser-induced ionization front propagating in air and plasma creation on the surface of a metal wire [24]. What is more, an optically multiplexed single-shot time-resolved probe can observe two-dimensional spatial imaging of intense laser pulse propagation dynamics, plasma formation, and laser beam filamentation [10]. However, most of these techniques rely on different optomechanical devices for pulse time separation, such as multiple mirror combinations and plane mirror splitting. Their imaging time resolution is usually several picoseconds, which limits the minimum time scale of their detection capability. Moreover, this strategy of separating pulses first and then combining requires a higher operating ability of the experimenter and the precision of optical instruments.

In this work, for observing the time evolution of plasma during femtosecond laser processing, a single-shot ultrafast multi-frame imaging system based on wavelength-polarization multiplexing is realized. Different from the pulse separation method in these imaging techniques [23,25,26], frequency doubling and polarization effects of nonlinear crystals are used to realize the timing control of the probe sub-pulses. The pulse sequence is coaxially propagated, and the images are acquired by different CCDs. These provide a reliable, stable and clear imaging observation. The spatial resolution reaches 228 lp/mm, and the temporal resolution is 200 fs, which is also higher than similar imaging techniques [10]. Other temporal resolutions can also be obtained by changing the optical components in the experiment. By observing femtosecond laser-induced plasma propagation, we revealed the mechanism of air ionization on early plasma shockwave expansion, as well as the multifilament propagation of the laser inside fused silica. However, it is worth considering that due to the influence of the coaxial propagation of the pulses, the nonlinear effects occurred during different wavelengths of pulses propagating in the crystal, which made the time resolution inconsistent with the calculation. Finally, different observations confirmed this phenomenon, and these also measured the time interval of each sub-pulse, which was 200 fs/1 ps.

## 2. System and Principle

Figure 1 shows a schematic of the experimental system. The femtosecond laser pulses are emitted by a Ti: sapphire chirped pulse-amplification system (Spectra-Physics Spitfire Ace, Spitfire Pro-35F1KXP, California, CA, USA). The repetition frequency is 1 kHz, and each pulse has a full width at half maximum of 35 fs and a center wavelength of 800 nm. The pulse is horizontal polarization. The pulses are subsequently divided into pump pulses and probe pulses using a beam splitter (Thorlabs, Inc., Newton, NJ, USA) at a splitting ratio of 3:7. The pump pulses are focused using a microscope objective lens (10×, NA = 0.25, Olympus, Tokyo, Japan) to irradiate a sample surface vertically. The probe pulses carrying sample information will also pass through a microscope objective lens (20×, NA = 0.4, Olympus) and finally are acquired by different CCD cameras (ImagingSource DMK 23U445; Ottobrunn, Germany) through spatial separation. The time interval between the probe pulse and the pump pulse is adjusted by a one-dimensional motorized translation stage (the minimum linear displacement distance is one μm, Zolix, Inc., Beijing, China) placed in the probe optical path. A continuous attenuator is placed in the pump path to adjust the laser energy. To avoid interference from the probe, we also place a half-wave plate to adjust the pump laser polarization angle to 45° horizontally before the laser enters the objective lens. In the probe path, to satisfy the coaxial propagation of pulses, we exploit the different nonlinear crystals to generate pulse sequences with specific time delays in time (Figure 1b). First, a horizontally polarized 800 nm femtosecond laser pulse passes through a BBO crystal (the thickness is 0.3 mm) to generate a 400 nm pulse. Here, we do not use any filters, so the outgoing lasers are 800 nm and 400 nm. In order to meet the phase matching principle, the angle between the polarization direction of the initial laser (800 nm) and the optical axis of the crystal is adjusted to 29.2° (the geometric schematic diagram is shown in Appendix A) [27], so that the 800 nm and 400 nm pulses can be emitted from the crystal at the same time.

Next, two probe pulses pass through a quartz birefringent crystal, and according to the principle of birefringence, the emitting pulses are 400 nm ordinary laser (400-o) and extraordinary laser (400-e), 800 nm ordinary laser (800-o) and extraordinarily laser (800-e). Then the Sellmeier equation is used to calculate [28,29] the refractive index $n_{o}$ and $n_{e}$ of each sub-pulse:(1)$$n_{o}^{2}=1.28604141+\frac{1.07044083\cdot \lambda^{2}}{\lambda^{2}-1.00585997\times 10^{-2}}+\frac{1.10202242\cdot \lambda^{2}}{\lambda^{2}-100}$$
(2)$$n_{e}^{2}=1.28851804+\frac{1.09509924\cdot \lambda^{2}}{\lambda^{2}-1.02101864\times 10^{-2}}+\frac{1.15662475\cdot \lambda^{2}}{\lambda^{2}-100}$$
where *λ* is the central wavelength of the laser, μm.

However, when a laser pulse travels through an anisotropic material, the characteristics of the laser pulse not only depend on the phase velocities but also on the group velocities. Due to the dispersion of laser pulses, the group velocity mismatch will inevitably lead to a change in propagation time. Therefore, the group velocity of sub-pulses should be used instead of phase velocity, and the formula is as follows [30]:(3)$$v_{g}=\frac{c}{n}{\left(1-\frac{\lambda }{n}\cdot \frac{\partial n}{\partial \lambda }\right)}^{-1}$$
where c is the speed of light in vacuum, 3.0 × 10^8^ m/s; n is the refractive index, $n_{o}$ or $\;n_{e}\;$.

Finally, the group refractive indexes $n_{g}$ of each sub-pulse are used to calculate the traveling time t through the quartz crystal:(4)$$n_{g}=c/v_{g}$$
(5)$$t=d\cdot n_{g}/c$$

Therefore, the group refractive index of 800-o, 800-e, 400-o, and 400-e are 1.5376, 1.5465, 1.5572, and 1.5668, respectively. In combination with the thickness of the crystal (6.4 mm) and the group refractive index, the traveling time of each sub-pulse in the crystal is $3.2802\times 10^{-11}$ s, $3.2992\times 10^{-11}$ s, $3.3220\times 10^{-11}$ s, and $3.3425\times 10^{-11}$ s, indicating that the time interval between two adjacent sub-pulses is approximately 200 fs. In addition, the spatial resolution test imaging of the system is shown in Figure 1c (Appendix A), which reaches 228 lp/mm by imaging the resolution test (228 pairs of lines per millimeter width, R2L2S1N1-NBS 1963A, Thorlabs, Newton, NJ, USA).

## 3. Results and Discussion

### 3.1. Time Evolution of a Femtosecond Laser-Induced Air Plasma Filament

Without placing the quartz crystal, we first observed the propagation of femtosecond laser-induced air plasma filaments using probe pulses at 400 nm and 800 nm wavelengths to verify that the two pulses emitted from the BBO were synchronized in time. As Figure 2 illustrates, we focused a pump laser pulse in the air through an objective lens (10×) to excite air molecules for ionizing and generating plasma filaments [31]. In the single-pulse excitation mode, the changes of the air filaments were observed with a laser fluence of $4.8\times 10^{15}\;W/{\mathrm{cm}}^{2}$. Under the 800-nm probe, we defined the 0.0 ps when the air filament occurred initially. At room temperature, the start time of impact ionization in the air is approximately 300–800 fs [32]. In the initial stage of ionization, air molecule ionization is dominated by multiphoton ionization and tunnel ionization. At 0.0 ps, because the laser energy was higher than that required for ionization of air molecules (more than $2.0\times 10^{12}\;W/{\mathrm{cm}}^{2}$ and less than $6.16\times 10^{13}\;W/{\mathrm{cm}}^{2}$), the air molecules were ionized, and the plasma occurred in front of the focused spot as predicted by Keldysh’s theory [33]. In addition, because of the low laser energy, the ionization form was mainly multiphoton ionization. As the probe delay increased, the laser focusing was greater than the defocusing effect caused by plasma, as well as the laser energy was much higher than that absorbed by the ionization of air molecules. Therefore, the laser continued to propagate to the focusing center, and as the laser intensity increased (more than $6.16\times 10^{13}\;W/{\mathrm{cm}}^{2}$), tunnel ionization played a dominant role in the plasma formation process. Currently, the intensity of the laser electric field is high enough to distort the Coulomb potential of atoms or molecules, resulting in electrons easily escaping from atoms. In addition, the self-focusing was always greater than the defocusing, and the filament length increased linearly. However, as the laser was continuously absorbed by air molecules, the laser self-focusing and plasma defocusing reached a dynamic balance, and the filament length reached saturation at 1.0 ps. It did not increase further due to the clamping effect of the laser intensity as the time delay increased.

This phenomenon also occurred under the 400-nm probe laser, as illustrated in Figure 2b, but the formation of the initial filament was not observed at 0.0 ps. This is due to the low plasma density in that early stage. The sensitivity of plasma detection was higher for longer wavelengths, leading to a time delay in the appearance of shadow images at 400 nm compared to the 800 nm probe. However, when the probe time delay reached 0.2 ps, the lengths of filaments were both 104 µm, showing that the two probe pulses of 400 nm and 800 nm had a temporal coincidence relationship.

Furthermore, we placed a quartz birefringent crystal with a thickness of 6.4 mm in the system to generate four sub-pulses. The ultrafast dynamics of the air filaments induced by the femtosecond laser were still observed. The filament excited by a pump pulse observed under the fourth sub-pulse in time (400-e) was used as a reference (see Appendix A). Because the prior three sub-pulses were more advanced in time, we could not observe any phenomenon at this moment. However, when the delay time increased, we discovered that the time interval of the four sub-pulses was not the same as theoretically calculated. For example, in Figure 3, we defined the 0.0 ps as the moment in the length of the filament observed by the 400-e reaching 105 µm. Currently, the first three sub-pulses did not detect the air plasma filament. As the delay time increased, the air filament was observed by the third sub-pulse (400-o) at 0.2 ps, with a length of 101 µm. This confirmed the results of previous theoretical calculations. However, until the time delay reached 1.0 ps, the filament was observed by the second sub-pulse (800-e), and the length reached 103 µm at 1.2 ps. In addition, the filament length under the first sub-pulse (800-o) was 112 µm at 1.4 ps. This phenomenon was different from the calculation. The time interval from 800-e to 800-o was approximately five times more than the theoretical expectation.

This was a slowing of the speed of light: that is, a slow light phenomenon caused by electromagnetically-induced transparency (EIT) as coaxial femtosecond pulses traveling in the crystal [34,35]. When the 800 nm and 400 nm femtosecond laser pulses entered the quartz crystal at the same time, due to the different group refractive index after dispersion, the speed of 800 nm was gradually higher than 400 nm during the pulse traveling. The atomic groups in the crystal were excited by the 800 nm pulse, transited from the ground state to the absorption band, then rapidly decayed to the metastable state, and eventually returned to the ground state. However, due to the excitation effect of 400 nm, the electron population oscillated between the ground state and metastable energy state. The 800 nm pulse could efficiently scatter off the temporally modulated ground state population into the 400 nm pulse, thus reducing the absorption of 400 nm, which leads to a rapid increase in the refractive index of 400 nm. Eventually, it leads to an increase in the time interval between the 400 nm and 800 nm emitted from the quartz crystal [36,37].

### 3.2. Ultrafast Observation of Early Shock Waves in the Femtosecond Laser Ablation of Fused Silica

In the following experiment, a side-polished fused silica sample (10 mm × 10 mm × 1 mm) was placed on a three-dimensional sample stage, and a pump laser pulse was focused on the side surface of the sample using the microscope objective lens (10×, NA = 0.45, Olympus Inc.). Subsequently, the shock wave generated by the femtosecond laser ablating the sample was observed [38], and the fluency of the pump laser was $1.2\times 10^{16}\;W/{\mathrm{cm}}^{2}$. According to the authors of [39], in the early period of the femtosecond laser ablation of materials, the types of femtosecond laser-induced shock waves comprised mainly an external hemispherical shock wave and a columnar airwave. The columnar airwave was by air molecules ionized, and the material shock wave occurred after 160 ps, then the rapid expansion was in hundreds of picoseconds. Because the time interval between sub-pulses in our system was in the hundreds of femtoseconds, early shock wave changes were observed. As shown in Figure 4, during the entire time window of observation, including imaging multiple times by one camera or simultaneous imaging by four cameras, we found that the variation in the expansion distance of the shock wave was very insignificant, and the expansion distance of the radial remained approximately 91 µm, which was inconsistent with previous observations. The hemispherical shock wave occurred at least one ps earlier than the ionization of air molecules. The main factor affecting this phenomenon was laser-induced air ionization. When the focused laser pulse arrived at the surface of the material, the front of the pulse was reflected by the surface, creating a superimposed light field. This caused the ionization threshold of the air near the surface to be reduced by two to three orders of magnitude, resulting in earlier ionization of air near the surface compared to slightly distant. This generated a large amount of air plasma, which was also reflected by the material surface, creating a hemispherical shock wave due to the pressure difference with the environment. Additionally, the free electrons ejected from material ionization increased the plasma density near the sample surface. Subsequently, the air molecules at a slight distance from the sample surface started to ionize, forming a columnar plasma channel. Since the interaction time between the femtosecond laser and the material only lasted about a hundred femtoseconds, there was no additional energy injection after the shock wave formation, and the shock wave morphology did not change significantly in a short time. In summary, the early shock wave was composed of air-ionized plasma, and its morphology was affected by the air breakdown. Moreover, since the material molecules were still in the energy-absorbing stage during such a short time, a large amount of plasma from the material was not ejected, and a small number of free electrons could not form a relatively obvious shock wave. As illustrated in Figure 4a–d, because the energy of the front of the pump had reached that required to ionize the material, material plasma was excited, which increased the plasma density near the surface of the material. Under the 800 nm probe pulse, the plasmon luminescence phenomenon appeared at the focus point of the pump pulse.

The 400-e was still a benchmark to define the 0.0 ps of the probe and pump laser. At this time, we observed the clear semicircular air shock wave and the cylindrical air plasma, though it was not obvious. The cylindrical air waveform was caused by femtosecond laser-induced air breakdown, while the other three sub-pulses could only capture a semicircular air shock wave. When the time delay was 0.2 ps, the cylindrical air plasma extended forward and interacted with the semicircular air shock wave, resulting in a small plasma coupling zone. Moreover, the same phenomenon was observed in the 400-o at 0.4 ps, the 800-e at 1.4 ps, and the 800-o at 1.6 ps. The time interval was the same as our previous observations of femtosecond laser-induced air filaments. In this way, the time interval between the four sub-pulses was determined to be 0.2 ps (a–b), 1.0 ps (b–c), and 0.2 ps (c–d). Therefore, we can realize the application in femtosecond processing by this imaging system through a single shot, and the observations at different time scales can also be captured because of the time interval difference.

### 3.3. Excitation and Propagation of Plasma Filaments in Fused Silica

In the third experiment, we made ultrafast observations of plasma filaments excited by a femtosecond laser in a transparent material [40] with a fluence of $4.8\times 10^{15}\;W/{\mathrm{cm}}^{2}$. The fused silica polished on three sides was placed on the sample stage, similar to the above experiment, taking 400-e as the probe-pump delay time reference (Figure 5a). At 0.0 ps, a large number of free electrons were excited in the fused silica due to the laser ablation (the same phenomenon also occurred in the 400-o at 0.2 ps, the 800-e at 1.2 ps, and the 800-o at 1.4 ps). As the probe delay increased, the density of plasma excited was so high that the refractive index increased in the longitudinal direction. This was equivalent to a positive lens guiding the laser to continue focusing inside and exciting the plasma. In this process, the energy of the pulse front was continuously absorbed by the target, causing the ionization of material molecules. When the high-energy pulse center arrived, a large number of free electrons were excited, and the diameter of the plasma in front was larger than that behind [41]. When the time delay increased to 0.8 ps, the density of free electrons began to decay significantly. As depicted in the column of Figure 5aIV, because of the continuous absorption of the laser energy by the material, the generated free electron density gradually decreased. In addition, in the early stage of material ionization, the self-focusing of the laser was always greater than the defocusing caused by the plasma. The laser moved forward during the dynamic equilibrium process (Figure 5c). Finally, the laser was focused on a certain area inside the material, with the focus position farther than that in the air [42].

In Figure 5b, the propagation distance of the plasma front was calculated, and it exhibited a linear growth trend with an increase in the probe delay. By linear fitting the image data obtained from the four sub-pulses, the slopes, namely the propagation velocities, were determined to be 1.90 × 10^8^ m/s, 1.95 × 10^8^ m/s, 1.92 × 10^8^ m/s, and 1.92 × 10^8^ m/s, which were close to the theoretical calculation (2.06 × 10^8^ m/s). Therefore, this also proved that the time interval of each sub-pulse could be measured by observing the plasma propagation, and the consistency of imaging between the four CCDs was also confirmed.

## 4. Conclusions

In this work, an ultrafast single-shot multi-frame imaging technique was used to observe the time evolution of femtosecond laser-induced plasma. The system was simple in principle and easy to set up to observe the femtosecond laser processing. The coaxially propagated pulse sequence ensured that the same position of the sample could be detected with high temporal resolution and spatial resolution. Other temporal resolution could also be obtained by changing the optical components in the experiment (see Appendix A). By observing the laser-induced plasma generation, propagating and expansion, we have performed ultrafast observations and mechanistic explanations of laser-induced air plasma filaments, early shock waves influenced by air ionization, and multifilament propagation of laser inside fused silica. Meanwhile, we found that the time intervals between probe sub-pulses and the calculation based on the refractive index were inconsistent, while the time interval between 800-e and 400-o had increased by almost five times. We think this was an electromagnetically-induced transparency (EIT) phenomenon caused by the coaxial propagation of double pulses, causing the beam of the 400 nm pulse to slow down when the 800 nm pulse propagates in the crystal. In the process of observing the expansion of the plasma shock wave, we discovered that an air shock wave did not change with time before the material shock wave expanded. As a result of the superimposed light field generated by the reflected laser pulse near the material’s surface, the ionization threshold of air molecules was increased by two to three orders of magnitude, leading to the earlier ionization of air molecules in close proximity to the surface compared to those further away. These pre-ionized air plasmas gave rise to the early shock waves, which exhibited a stable morphology over a relatively short time due to the absence of any additional energy injection. We also observed the multi filament propagation of plasma infused quartz crystal induced by a femtosecond laser. In conclusion, by capturing the same experimental phenomenon in different experiments, such as the position of the plasma front inside the crystal induced by the pump pulse observed by four probe sub-pulses, we indirectly measured the temporal resolution of the system. The time intervals were 0.2 ps between 800-o and 800-e, 1.0 ps between 800-e and 400-o, and 0.2 ps between 400-o and 400-e. This feedback on the time resolution of the system through the experimental results also provides a reference for the implementation of the follow-up observation technology.

## Figures and Tables

**Figure 1 materials-16-03264-f001:**
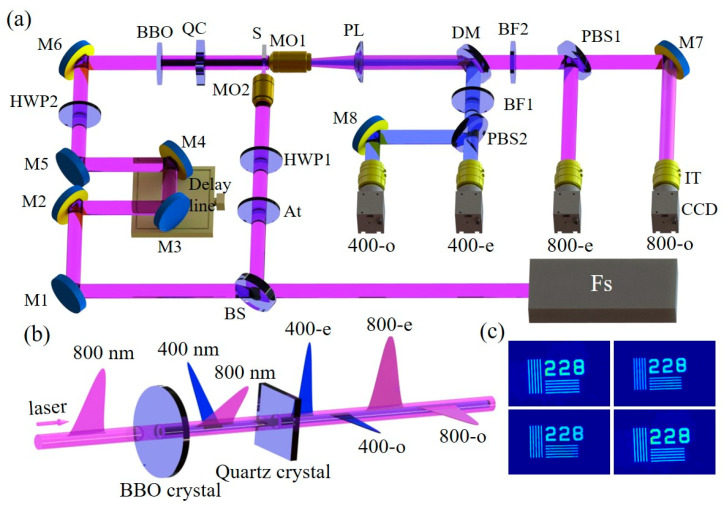
(**a**) Experimental setup for the single-shot ultrafast multi-frame imaging. The red beam represents 800 nm while the blue is 400 nm (**b**) Probe pulse train propagation. (**c**) Spatial resolution test under four sub-pulses, 228 lp/mm, the length of each line is 52.6 μm, and the width is 2.2 μm. Note: M1–M8: Reflect mirror; BS: Beam splitter; BBO: β-barium borate crystal; QC: Quartz crystal; MO: Microscope objective; S: Sample; DM: Dichroic mirror; PBS1: Polarizing beam splitter (800 nm); PBS2: Polarizing beam splitter (400 nm); At: Attenuator; HWP1-2: Half-wave plate; PL: Plano-convex lens; CCD: Charge-coupled device camera; IT: Imaging tube; BF1: Bandpass filter (400 nm); BF2: Bandpass filter (800 nm).

**Figure 2 materials-16-03264-f002:**
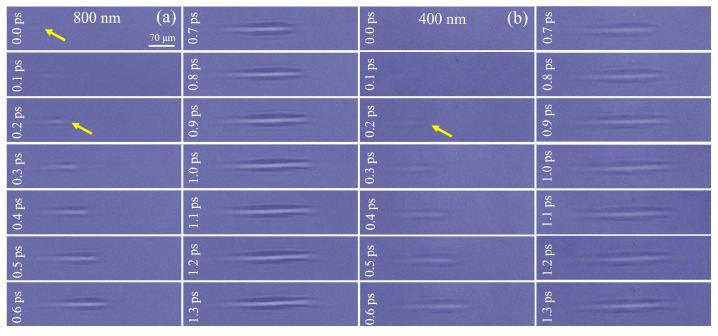
Time evolution of a femtosecond pulse laser-induced air plasma filament under (**a**) an 800-nm probe pulse and (**b**) a 400-nm probe pulse. The yellow arrow indicates the initial formation of air plasma filament.

**Figure 3 materials-16-03264-f003:**
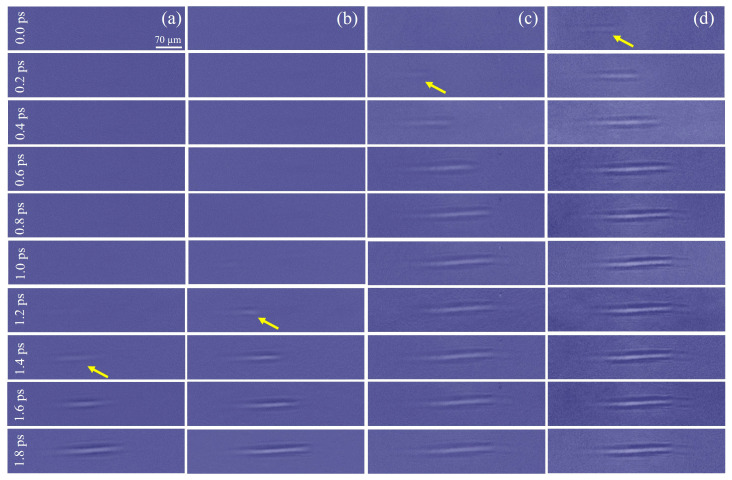
Time evolution of a femtosecond pulse laser-induced air plasma filament under different probe pulses at (**a**) 800-o; (**b**) 800-e; (**c**) 400-o, and (**d**) 400-e. The yellow arrow indicates the initial formation of air plasma filament.

**Figure 4 materials-16-03264-f004:**
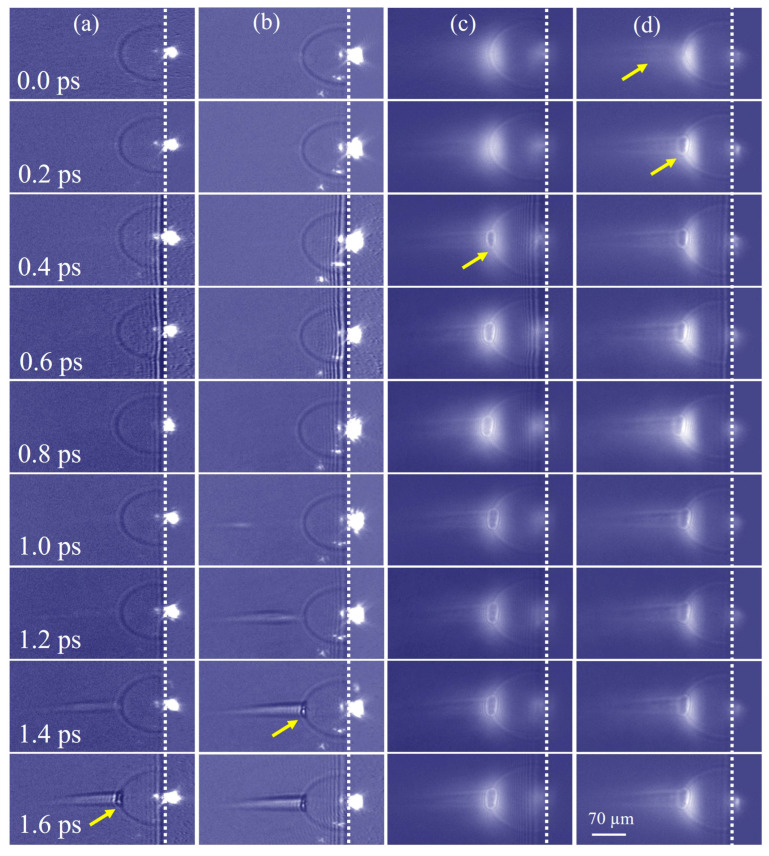
Time evolution of shock wave generated by a femtosecond pulse laser ablation on the surface of fused silica under (**a**) 800-o; (**b**) 800-e; (**c**) 400-o, and (**d**) 400-e probe pulse. The white dotted line is the interface between air and sample. The first yellow arrow in column (**d**) indicates the formation of an air ionization channel. The other yellow arrows indicate the area of interaction between air ionized plasma and the air shock wave.

**Figure 5 materials-16-03264-f005:**
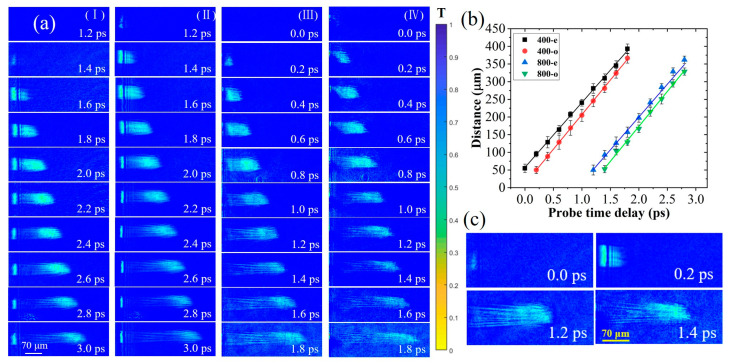
Time evolution of plasma filaments in fused silica induced by a femtosecond pulse laser. (**a**) Plasma images under four sub-pulses at (**I**) 800-o, (**II**) 800-e, (**III**) 400-o, and (**IV**) 400-e. The 0.0 ps was the time delay between the 400-e and the pump pulse. The images within 1.2 ps of the 800-o and 800-e were omitted because they did not detect any plasma information. (**b**) The propagation distance of the plasma front is observed by the four sub-pulses. (**c**) Plasma propagation is observed by a single shot.

## Data Availability

The data presented in this study are available on request from the corresponding author.

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
