# Peer review of "Single-Shot Multi-Frame Imaging of Femtosecond Laser-Induced Plasma Propagation"

_materials, 2023, doi:10.3390/ma16083264_

Round 1

Reviewer 1 Report

Authors report an interesting methodology to image femtosecond laser-induced 2 plasma propagation. I think the subject matter is of considerable contemporary interest and authors presented their results neatly. In view of this, this paper can be accepted for publication by adding following minor revisions:

1. Femtosecond laser induced plasma lifetime spans from few femtoseconds. However, authors reported the same from picoseconds  only. What could be the reason rather what is the limiting factor (ps not fs)?

2. Figure 5 (b): why no error bars?  

Reviewer 2 Report

The paper “materials-2290409” related to laser cladding was reviewed. Please follow the comments carefully and resubmit your paper for the next consideration and reviewing process.

1.     Please add a brief statement on your methodology in the abstract.

2.     Add brief quantitative results to the abstract

3.     The size of the wording in “Figure 1. (a) Experimental setup for the single-sh” is too small.

4.     The explanation in Figure 4 “Time evolution of shock wave generated” needs to be expanded.

5.      Laser has many advantages over the conventional manufacturing method which can be highlighted in your paper. Please read the following manuscript and add it to the literature to show how the laser is comparable with conventional manufacturing. “Laser subtractive and laser powder bed fusion of metals: review of process and production features”

6.      List the detail of the findings in your conclusion.

Reviewer 3 Report

Tianyong Zhang et al presented a single-shot ultrafast multi-frame imaging system based on wavelength-polarization multiplexing. It allowed plasma imaging of four time-delay separated images, with non-equal delays. In general, authors suggest a relatively simple and cheap for realization method of fast 4-image framing.

I would suggest this work for publication as it is. Only, I address one question to authors: Did you try quartz crystal with different thickness to confirm your conclusions about "This was a slowing of the speed of light, means that, a slow light phenomenon caused by electromagnetically induced transparency (EIT) as coaxial femtosecond pulses traveling in the crystal [33]"?

Round 2

Reviewer 2 Report

This manuscript is ready to publish.